# Scale-Up and Testing of Polyurethane Bio-Foams as Potential Cryogenic Insulation Materials

**DOI:** 10.3390/ma15103469

**Published:** 2022-05-12

**Authors:** Maria Kurańska, Ugis Cabulis, Aleksander Prociak, Krzysztof Polaczek, Katarzyna Uram, Mikelis Kirpluks

**Affiliations:** 1Faculty of Chemical Engineering and Technology, Cracow University of Technology, Warszawska 24, 31-155 Cracow, Poland; maria.kuranska@pk.edu.pl (M.K.); krzysztof.polaczek@doktorant.pk.edu.pl (K.P.); katarzyna.uram@doktorant.pk.edu.pl (K.U.); 2Polymer Laboratory, Latvian State Institute of Wood Chemistry, Dzerbenes 27, LV-1006 Riga, Latvia; mikelis.kirpluks@kki.lv

**Keywords:** scale-up PUR bio-foam, bio-polyol, spray bio-foam, cryogenic insulation

## Abstract

This article compares the properties of closed-cell PUR bio-foams produced on a laboratory scale and on an industrial scale. In the formulation used, the polyol premix contained 40 wt.% of a bio-polyol based on rapeseed oil. Selected useful properties of the foams obtained on the two scales and the use of one-step and spraying methods were compared. In the case of the spraying method, the experimental system was compared to a commercial one. Given the possibility of applying the bio-foams in insulation systems for cryogenic and liquefied natural gas (LNG) applications, a compressive strength analysis of the foams was carried out at room temperature as well as at −196 °C. It was found that the foams modified with the bio-polyol were characterized by a higher compressive strength at low temperatures than commercial foams based on a petrochemical polyol.

## 1. Introduction

Commercially-available spray polyurethane (PUR) foams are usually prepared from petroleum-based components such as polyether or polyester polyols and polyisocyanates. Such foams make up one of the fastest-growing building insulation material markets [1,2]. The cell structure type (open- or closed-cell) of spray foams has an influence on their potential application. Open-cell spray foams are applied to the attics of buildings given their low water vapor diffusion resistance factor and low apparent density (8–15 kg/m^3^). Closed-cell spray PUR foams, on the other hand, are characterized by lower thermal conductivity (0.021–0.028 W/m·K) [3] and higher apparent density (34–60 kg/m^3^) [4] than open-cell foams (0.035–0.042 W/m·K) [5].

The synthesis of polymers from renewable resources has been investigated by leading research teams from different countries for more than 40 years; the first review articles about this topic were published in the early 2000s [6,7,8]. Taking into account the newest trends in the development of polymeric materials, including PUR foams, in addition to their modification using renewable raw and recycled materials, a partial or full replacement of petrochemical polyols with bio-polyols is preferred [9,10,11,12,13,14,15,16]. Kurańska et al. have studied rigid PUR foams produced with used cooking oil. In their work, bio-polyols with different hydroxyl values and viscosities were used to prepare open-cell PUR foams. Beneficial effects of the bio-polyols on both the cellular structure and the mechanical properties and thermal conductivity of the modified foams were observed [17]. In the research of Kairyte et. al., a polyethylene terephthalate waste-based polyol and a sucrose-based polyol were used in order to obtain dimensionally-stable PUR foams [18]. Pan and Saddler analysed the effect of replacing a petrochemical polyol with polyols obtained from organosolv and kraft lignin on the properties and structure of a rigid PUR foam [19]. Eco-polyols can be also applied in the synthesis of polyurethane-polyisocyanurate (PUR-PIR) foams [20,21]. Borowicz et al. analyzed the influence of an eco-polyol based on waste polylactide on the properties of rigid PUR-PIR foams. They concluded that the PUR-PIR foams modified by the eco-polyol had better functional properties than the reference foams based on a petrochemical polyol [22,23]. 

Through the chemical modification of vegetable oils, it is possible to obtain new compounds characterized by different chemical structures. There are many methods of chemically modifying of vegetable oils, including the epoxidation and opening of oxirane rings and transesterification [24,25,26,27].

Spray insulation PUR systems have so far been considered to be the most effective and fastest-developing thermal insulation technology. PUR foams are used primarily in building construction and are suitable for the thermal insulation of almost all structural elements. In addition, spray insulation has many other applications within the construction industry. It is highly effective in agriculture, where it serves in the insulation of cold stores, livestock facilities, and fruit and vegetable storage facilities. Spray foams can also be used in the transport industry, in the thermal insulation of tanks storing liquefied gases at low temperatures. 

PUR foam systems are applied by spraying onto an insulated surface using high pressure dispensing equipment in a 1:1 (by volume) dosage ratio. The literature offers the results of research on PUR foams modified with bio-polyols of various chemical structures and their impact on the foaming process, as well as analyses of their functional properties. Most studies concern the influence of the properties of bio-polyols and their contents on the properties of PUR foams obtained on a laboratory scale. So far, little attention has been paid to scaling up the production of PUR bio-foams intended for use as thermoinsulating spray materials.

This paper presents the properties of foams obtained on both a laboratory scale and an industrial one, with the use of a high-pressure device. The properties of the foams were also compared in relation to the method used for their production. In addition, the properties of the spray foams were compared to the reference foam developed in earlier studies, as well as to the foams obtained for use with commercially available systems [27,28]. In the reference material, 40% of a petrochemical bio-polyol was replaced with a rapeseed oil-based bio-polyol.

## 2. Results and Discussion

Spray PUR foams with closed-cell structures are an important part of heat insulating solutions as they provide a technique that can be applied in situ. The adaptability of spray PUR systems combined with the wide range of apparent densities suitable for different applications offers versatility for in-situ application. Closed-cell PUR foams can be applied using various thicknesses and apparent densities depending on the insulation requirements. 

Cellular structure is one the most important parameters of porous PUR materials. Cell size, wall thickness and anisotropy have a significant impact on the physical and mechanical properties of PUR foams [29]. What is more, the cellular structure of foams is affected by the properties of the components used in their synthesis; this can affect density, viscosity, reactivity, as well as the conditions of the foaming process. 

In this study, the influence of changes in the PUR formulation as well as the method of PUR system application on the properties of foamed materials was analyzed. The reduction of the isocyanate index of 1:1.1 in the BIO_PUR material was performed to ensure a 1:1 volume ratio in the BIO_PUR_1:1 material as such a formulation can be tested on an industrial scale due to the possible work conditions of the high-pressure spraying machine. Then, the BIO_PUR_1:1 system was tested on an industrial scale, obtaining the BIO_PUR_ spray material. It allowed for the analysis of the influence of the foaming method on the properties of PUR bio-foams, and for the comparison to the foamed material obtained on the basis of a commercial system, using the same high-pressure machine. 

The foaming process is a key step in the preparation of porous PUR materials. During this step, the aforementioned cellular structure is generated. An analysis of the foaming process was completed for the systems with the same formulations of polyol premix and different amounts of isocyanate. Figure 1 shows the changes of the dielectric polarization (a), temperature (b) and pressure (c) during the foaming process.

It was found that a slight change in the amount of isocyanate, which ensured the appropriate volume ratio during industrial tests, had no effect on the reactivity of the system. This is a very promising result as it shows no complications in the scaling up of the production of PUR bio-foams.

The microstructures of PUR materials in cross-sections parallel and perpendicular to the foam rise direction are shown in Table 1.

The lower value of the isocyanate index of the BIO_PUR_1:1 foam compared to the BIO_PUR foam resulted in a slight reduction in the size of the cells. Significant changes in the cellular structure of the foams were observed when the foaming method and scale were changed. The use of a high-pressure spraying device resulted in materials with much smaller cells (BIO_PUR_spray). That was due to the perfect mixing of the components (polyol premix and isocyanate). The commercial system was characterized by the smallest cell size. The microphotographs in Figure 2 show the border between two layers of the foamed materials obtained from the bio-based system and a commercial one. It can be seen that the cells of the commercial foam are characterized by much thinner walls which is related to its lower apparent density compared to the BIO_PUR_spray system (Table 2).

The cell density values of the rigid PUR foams with rapeseed oil-based polyol were determined and compared to the values typical of commercial products. The cell density in the cross-section perpendicular to the direction of foam growth was higher than in the parallel cross-section. The cross section perpendicular to the foam growth direction reveals a higher number of cells with a smaller surface area (Table 2) compared to the structure obtained for the parallel cross-section as shown in the images in Table 1. The BIO_PUR_spray foam had higher cell density values with respect to the foams obtained on a laboratory scale. The spray PUR foam with the bio-polyol has a reduced cell density compared to commercially available foam material. The higher cell density may have a beneficial effect on the thermal conductivity of a commercial foam (Figure 3).

The apparent density of PUR foams is closely related to their thermal insulation (Figure 3) and mechanical properties as well as their dimensional stability. Increasing the synthesis scale of the BIO_PUR_1:1 foam resulted in a reduction of its apparent density. From an economic point of view, it is advantageous if the foams have sufficient dimensional stability. PUR spray foams are used as thermal insulation materials; therefore, a low thermal conductivity coefficient is one of their most important properties. Among the foams modified with the bio-polyol, the material BIO_PUR_spray was characterized by its favorable thermal insulation properties. Compared to the BIO_PUR_1:1 foam having the same formula and obtained in laboratory conditions, the thermal conductivity coefficient of BIO_PUR_spray was lower by 1 mW/m·K. The commercial foam had the lowest thermal conductivity coefficient. However, it should be taken into account that in the selected commercial formulation a physical blowing agent was also used as a foaming agent. In accordance with the principles of green chemistry, one should strive to introduce environmentally friendly components. Foams containing both a bio-polyol as well as a chemical blowing agent meet this idea.

The compressive strength of the PUR foams was tested in directions parallel and perpendicular to the foams rise direction at 10% relative strain at 294 K (21 °C) and at 77 K (−196 °C) (Figure 4 and Figure 5). Figure 4b shows the normalized compressive strength (σ_norm_) of the foams. In this case, compression strengths were calculated with respect to the foam’s apparent density. The equation used for the normalization of strength is as follows [30]:(1)$$\sigma_{norm}=\sigma_{i}{\left(\frac{\rho_{average}}{\rho_{i}}\right)}^{2.1}$$

Where: $\sigma_{i}$—the experimental strength of given foam, $\rho_{i}$—the apparent density of the same foam, $\rho_{average}$ the average apparent density (40.6 kg/m^3^) calculated on the base of compared foams densities.

As can be seen in Figure 4a,b, the foams were characterized by differences in strength depending on the direction of testing, which is a direct result of their anisotropic cell structures. The highest compressive strength was observed for the BIO_PUR foam, which is due to the applied system with isocyanate index 1.1 in relation to the content of the polyol and water. The spray system obtained under laboratory conditions (BIO_PUR_1:1), had a lower compressive strength, resulting from a lower cross-linking density of the polymer matrix caused by a lower isocyanate index, as well as a lower apparent density. Application of the industrial spraying method, which significantly influenced the formation of cells with small dimensions and caused a decrease in apparent density, also resulted in a slight decrease in compressive strength. The commercial foam had similar compressive strength compared to the foams made from the bio-polyol HF. 

The polymer matrix of PUR foams is composed of two segments: hard and soft [31]. The physical properties of PUR foams depend largely on the chemical structure of the soft and hard segments, their shares in terms of mass, as well as on the structure formed in the process of phase separation. The rigid segments consisting mainly of highly polarized urethane bonds and the hard glass phase formed by them determine the mechanical properties of the material such as strength and hardness. At room temperature, the soft segments appear as a rubber phase. At liquid nitrogen temperature, the soft segment undergoes brittle crystallization through phase transformation, resulting in a significant increase in the stiffness and mechanical strength of the material [32]. The compressive test at −196 °C was carried out for the BIO_PUR_spray and commercial foams. Figure 5 shows the mechanical compressive strength in a direction parallel (Figure 5a) and perpendicular (Figure 5b) to the foam growth direction. At −196 °C, the foams exhibit significantly (three to even six times) higher compressive strength compared to the foams tested at room temperature. Comparing the normalized values of the mechanical strength of the BIO_PUR_spray and commercial foams, it was unexpectedly observed that at low temperatures, much more favorable changes occurred in the case of the developed materials. This is especially noticeable for the direction perpendicular to the foam growth direction, which is critical in taking into account the dimensional stability of the foamed materials. 

## 3. Materials and Methods

### 3.1. Materials

Rigid PUR foams were prepared using two types of polyols. The first was Rokopol RF551 (PCC Rokita Brzeg Dolny, Poland) and second was a bio-polyol HF (synthetized in the laboratory of Cracow University of Technology, Cracow, Poland). The bio-polyol HF was synthesized using a two-step method: epoxidation and opening of oxirane rings with 1.6-heksanediol (Figure 6). The characteristics of both polyols used to prepare the BIO_PUR foams are shown in Table 3.

An isocyanate compound (polymeric methylene diphenyldiisocyanate—pMDI), consisting of 31% of free isocyanate groups was supplied by Minova Ekochem S.A. (Siemaniowice Śląskie, Poland). A catalyst, amine-based compound that accelerates the gelation and foaming reaction was supplied by Evonik Industries AG (Essen, Germany). The same company provided a surfactant based on an organosilicon compound (Niax silicone LV-6915). Distilled water was used to create the porous foam structure, which reacted with the isocyanate to produce carbon dioxide. In addition, triethyl phosphate (TEP) supplied by Purinova Sp. z o.o. (Bydgoszcz, Poland) was used as a flame retardant.

### 3.2. Preparation of Rigid PUR Foams

Rigid PUR foams were prepared on a laboratory scale by a one-step method according to the formulations presented in Table 4 (BIO_PUR and BIO_PUR_1:1). Firstly, a polyol premix consisting of a petrochemical polyol, bio-polyol, catalyst, surfactant and water was prepared (Table 4). The polyols mixture contained 40 wt.% of the HF bio-polyol. The mixture was stirred for 60 s. Then an appropriate amount of isocyanate was added to the polyol premix, stirred for 3 s and poured into the mold. 

Generally, on an industrial scale, polyurethane foam is produced as a result of a chemical reaction consisting of the joining of two liquid components directly, at the outlet of the spray gun nozzle. Both components are delivered pneumatically to the installation site by means of high-pressure hoses in a heat shield. The main components of a spraying polyurethane system are two liquid components—polyol premix and isocyanate. The components are mostly mixed in a 1:1 ratio by volume. The components are delivered in barrels and, after mixing through the spray nozzles of the gun, they are applied in the form of a delicate spray to an insulated object (Figure 7). Spraying applications were conducted using a Wintermann 200 Pro machine manufactured by TBH POLSKA Sp. z o.o., (Sieroslaw, Poland) according to the formulation of BIO_PUR_spray (the same as for BIO_PUR_1:1 which was prepared in laboratory scale). In the case of the spraying method, the temperature of the components was higher than components in the case of laboratory method. Components’ temperature–spraying method: 40 °C; temperature of the mixing head: 40 °C; ambient temperature: 15 °C; components’ pressure 80 bar. In the case of the laboratory method, the temperature of the components was 20 °C.

The rigid polyurethane foams were conditioned for 24 h at room temperature before being cut into samples used later in measurements. 

### 3.3. Testing Methods

The foaming process was performed using a FOAMAT device (Format-Messtechnik GmbH, Karlsruhe, Germany). The device was connected to a computer and operated through a dedicated piece of software. Measurement provided information about the changes of dielectric polarization, which showed the reactivity of the polyurethane system. In addition, the device was equipped with a thermocouple, which measured the temperature inside the core of the growing rigid polyurethane foam.

The cell morphology was analysed using an optical microscope (PZO Warszawa). Anisotropy coefficient was calculated as the ratio of the foam cell height to cell width. All samples for structure tests were taken from the center of the foam, the so-called core. The cell density (*N_f_*) was determined according to Equation (2) [33]:(2)$$N_{f}={\left(\frac{N}{A}\right)}^{\frac{3}{2}}$$
where: *N* is the number of bubbles in a micrograph with the area *A* in cm^2^. 

The apparent density of the foams was calculated as the mass-to-volume ratio of a sample. The dimensions of the foam were 200 × 200 × 50 mm^3^ and the test was prepared according to ISO 845 standard. The same samples were used to measure thermal conductivity using a Laser Comp flow meter constructed in line with ISO 8301:1991. The measurements were taken at an average temperature of 10 °C (the cold plate temperature 0 °C and the warm plate temperature 20 °C). The closed-cell content in the rigid polyurethane foams was measured by the pycnometer method according to the ISO 4590:2016 standard. One measurement was performed using 12 foam cubes (25 mm edge length) and 24 oak cubes (20 mm edge length), which were tumbled together for 10 min at 60 rpm. A compressive strength test at 10% of deformation was completed in accordance with ISO 826. The compressive strength of the foams was measured in two directions, parallel and perpendicular to the rise direction of the foams using a Zwick Z005 TH Allround-Line instrument. The temperature of the measurement was 293K (room temperature). Apart from that, the same test was carried out by testing the samples at 77K (cryogenic temperature). In this case, a Zwick Z100 (Zwick GmbH & Co, Ulm, Germany) with special equipment (Figure 8) was used. Cylindrical samples of the foam with a height of 22 mm and a diameter of 20 mm were cut from the core of the foam block. Six foam samples were tested in each direction. Before testing, the samples were placed in a cryostat filled with liquid nitrogen for at least 5 min. 

## Figures and Tables

**Figure 1 materials-15-03469-f001:**
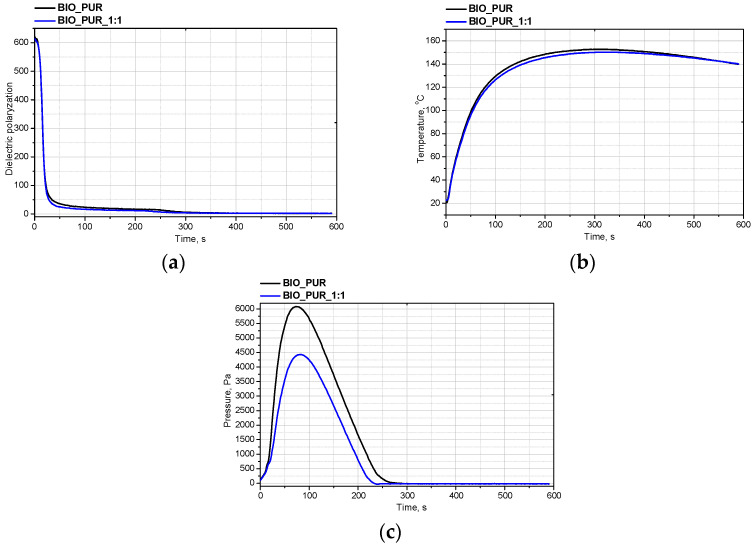
Influence of isocyanate amount on the dielectric polarization (**a**), temperature (**b**) and pressure (**c**) during the foaming process.

**Figure 2 materials-15-03469-f002:**
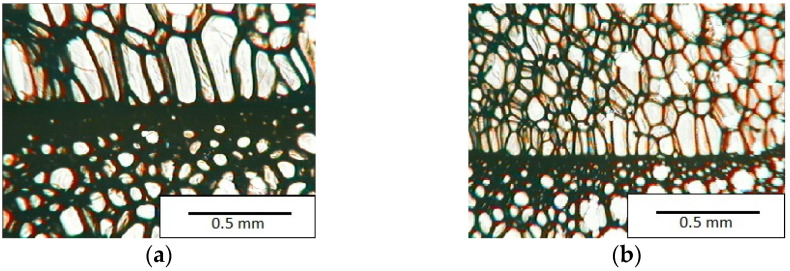
The border between two layers of foamed materials in the bio-based (**a**) and commercial systems (**b**).

**Figure 3 materials-15-03469-f003:**
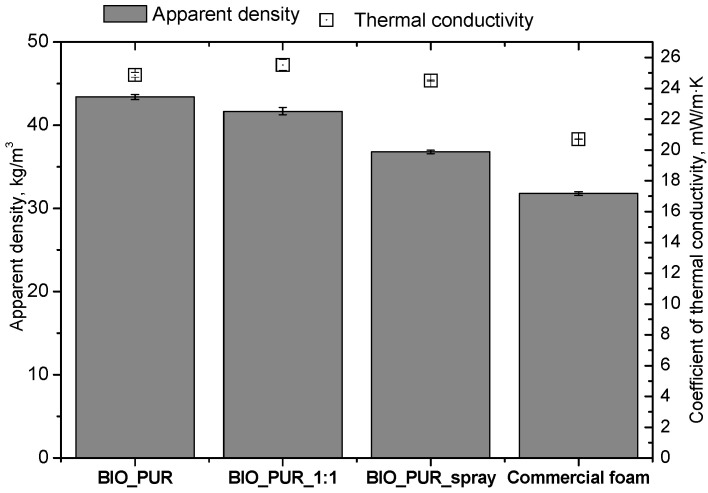
The apparent density and thermal conductivity of spray PUR bio-foams obtained on laboratory and industrial scales.

**Figure 4 materials-15-03469-f004:**
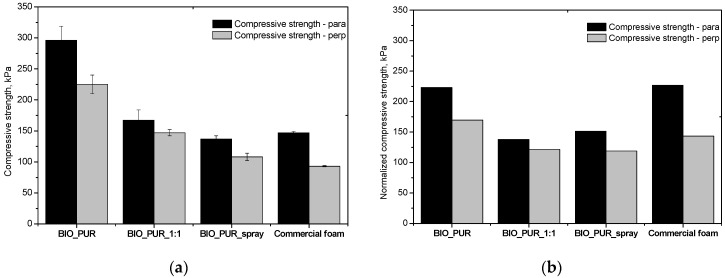
The compressive strength (**a**) and normalized compressive strength (**b**) of spray PUR bio-foams obtained in laboratory and industrial conditions. Scale measured in two directions: perpendicular and parallel to the foam rise direction.

**Figure 5 materials-15-03469-f005:**
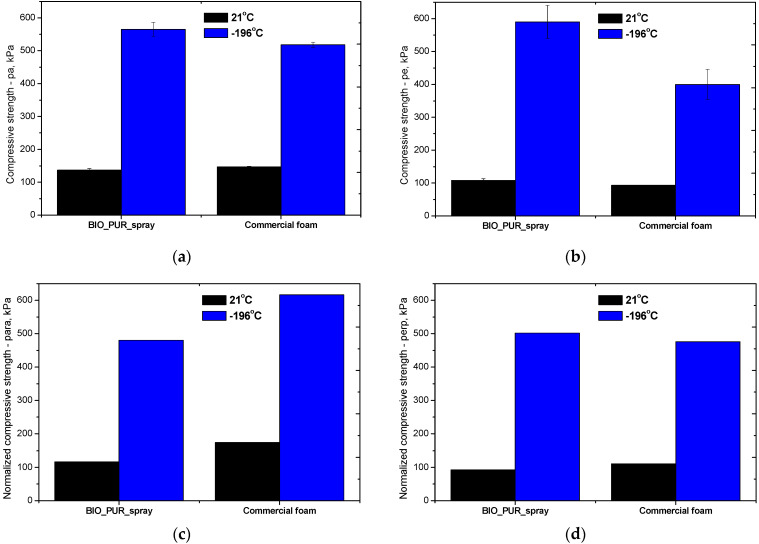
The compressive strength (**a**,**b**) and normalized compressive strength (**c**,**d**) of spray PUR bio-foams obtained on an industrial scale measured in two directions: perpendicular and parallel to the foam rise direction.

**Figure 6 materials-15-03469-f006:**
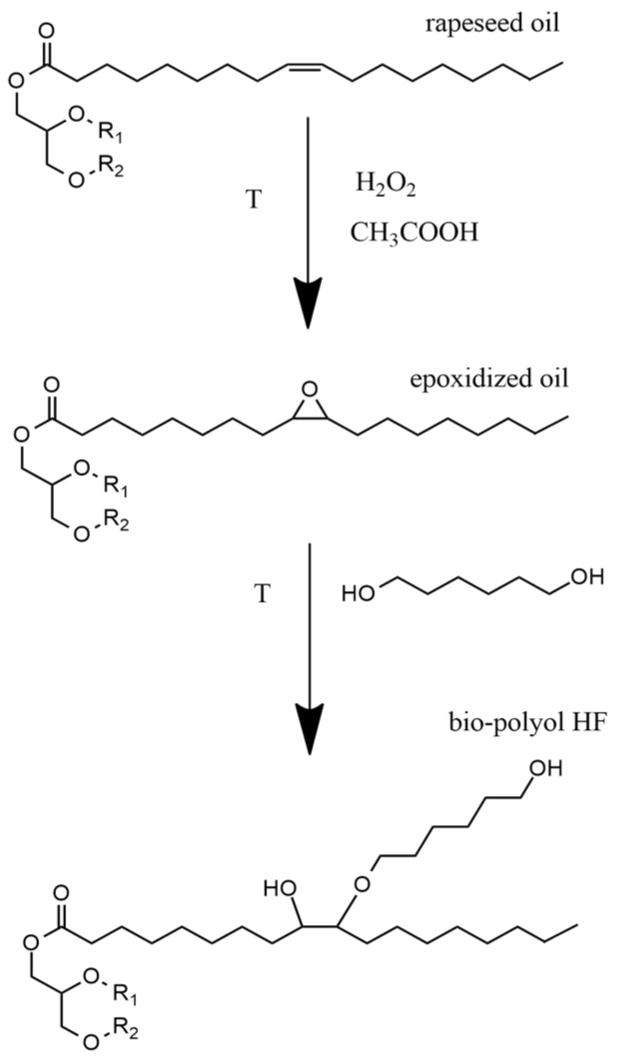
Scheme of epoxidation and opening of oxirane rings with 1.6-heksanediol (T = 80–90 °C).

**Figure 7 materials-15-03469-f007:**
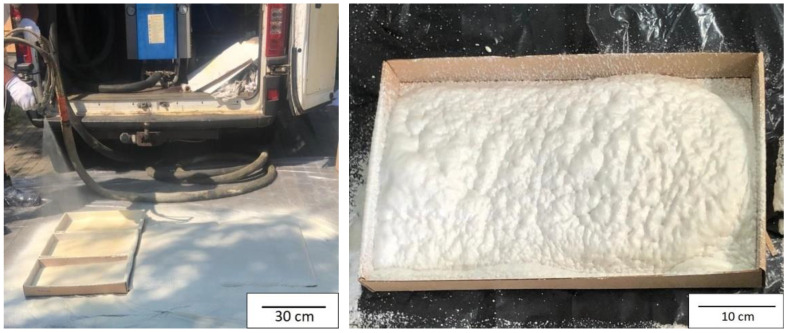
Samples prepared on an industrial scale.

**Figure 8 materials-15-03469-f008:**
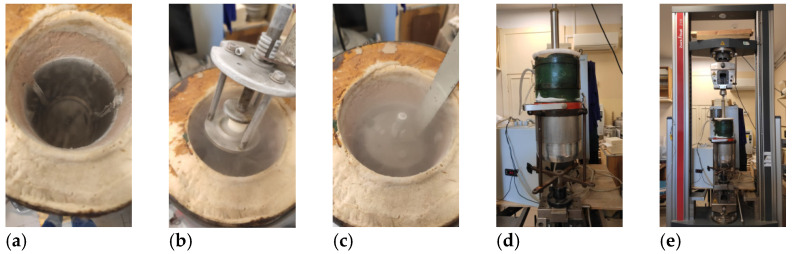
Equipment for polyurethane foam tests at cryogenic temperatures: (**a**) cryostat; (**b**,**c**) placing the sample in the cryostat; (**d**) closed cryostat; (**e**) cryostat connected to Zwick equipment for compression test.

**Table 1 materials-15-03469-t001:** The microstructure of PUR bio-foams.

Growth Direction	BIO_PUR	BIO_PUR_1:1	BIO_PUR_Spray	Commercial Foam
Parallel (para)	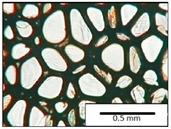	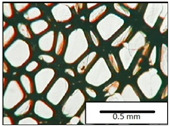	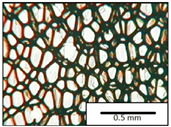	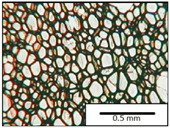
Perpendicular (perp)	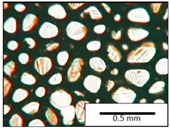	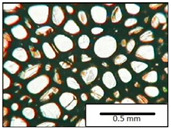	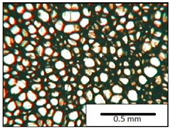	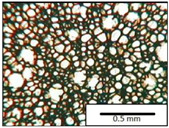

**Table 2 materials-15-03469-t002:** Structural parameters of PUR bio-foams.

Name of Sample	Anisotropy Coefficient		Cross Section Area 10^2^, mm^2^		Cell Density, 10^5^ cm^2^		Content of Closed Cells, %
	para	perp	para	perp	para	perp	
BIO_PUR	1.14 ± 0.06	0.89 ± 0.05	1.7 ± 0.2	1.3 ± 0.3	1.1 ± 0.1	1.79 ± 0.7	90.2 ± 1.8
BIO_PUR_1:1	1.27 ± 0.14	0.93 ± 0.02	1.6 ± 0.1	0.8 ± 0.1	1.3 ± 0.2	3.1 ± 0.5	90.6 ± 1.4
BIO_PUR_spray	1.32 ± 0.04	0.92 ± 0.03	0.5 ± 0.1	0.4 ± 0.1	7.8 ± 0.6	11.0 ± 2.1	88.3 ± 1.3
Commercial foams	1.31 ± 0.08	0.92 ± 0.06	0.4 ± 0.1	0.26 ± 0	18 ± 1.66	21.2 ± 1.0	89.0 ± 0.4

**Table 3 materials-15-03469-t003:** Characteristics of RF551 and bio-polyol HF.

Properties	RF551	Bio-Polyol HF
Hydroxyl value, mgKOH/g	428	250
Acid value, mgKOH/g	0.10	4.05
Water content, %mas.	0.10	0.38
Average molecular weight, g/mol	625	978
Viscosity, mPas	4000	5128
Functionality	4.77	4.36

**Table 4 materials-15-03469-t004:** Formulations of foam materials, in parts by weight (pbw).

Component, pbw	BIO_PUR	BIO_PUR_1:1/BIO_PUR_Spray
RF551	60	60
Bio-polyol	40	40
L6915	1.5	1.5
Polycat9	5.4	5.4
Water	3.4	3.4
TEP	30	30
PMDI	149.8	166.5

## Data Availability

The data presented in this study are available on request from the corresponding author.

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
