# Peer review of "Scale-Up and Testing of Polyurethane Bio-Foams as Potential Cryogenic Insulation Materials"

_materials, 2022, doi:10.3390/ma15103469_

Round 1

Reviewer 1 Report

This manuscript reports the preparation of polyurethane foams on industrial scale by spraying method, and the bio-polyol were used for modification to get higher compressive strength. May eventually be publishable, but requires major revisions as indicated.

There are quite a few issues that are not clear, which is important.

  1. There is no scale bar in the optical images of Table 1, Figure 2, 7, and 8. The scanning electron microscope (SEM) should be used to characterize the microstructure and morphology of PUR foams.
  2. Industrial production of PUR foams by spraying method should be supplied the detailed experimental steps.
  3. Some information should be contained in the optical figures. For example, the samples and equipment in figure 8 should be marked.
  4. Why the bio-polyol modification (rapeseed oil-based) can improve the strength? The bio-properties of PUR foams are suggested to supplied.

Author Response

Thanks for the reviews. The authors tried to answer all your questions and the article was supplemented with additional information

Rev.1

This manuscript reports the preparation of polyurethane foams on industrial scale by spraying method, and the bio-polyol were used for modification to get higher compressive strength. May eventually be publishable, but requires major revisions as indicated.

There are quite a few issues that are not clear, which is important.

  1. There is no scale bar in the optical images of Table 1, Figure 2, 7, and 8. The scanning electron microscope (SEM) should be used to characterize the microstructure and morphology of PUR foams.

The scale has been added in the drawings. Optical microscope analysis was chosen in place of SEM because a low magnification is enough to show the cell structure. Moreover,  the Aphelion image analysis software allowed  the exact size and anisotropy of cells to be determined.

  1. Industrial production of PUR foams by spraying method should be supplied the detailed experimental steps.

This part has been completed.

  1. Some information should be contained in the optical figures. For example, the samples and equipment in figure 8 should be marked.

More info is added in Fig.8 caption.

  1. Why the bio-polyol modification (rapeseed oil-based) can improve the strength? The bio-properties of PUR foams are suggested to supplied.

The mechanical properties at cryotemperature can be improved due to the oil structure of the biopolyols and hydroxyl grup positions. From one side shorter distance between urethane bonds, from other one a plasticization effect by dangling chains.

Reviewer 2 Report

General comments
Not suitable for publication in the present state. Reject and Resubmit

Specific comments
Abstract: Authors state "The article compares the properties of closed-cell PUR bio-foams produced on a laboratory scale and on an industrial scale" but the polyol sources used in laboratory PU foam and industrial PU foam is not the same. It's like comparing oranges with lemons though  both are in the citrus family. 
Introduction is very badly written and the cited papers are inappropriate. Authors say "full replacement of petrochemical polyols with bio-polyols is
preferred [6–13]" and cite papers published in 2020, 2021 etc. Seriously do authors think that until 2020s no one else thought of replacing petroleum based polyols with biobased polyols? Also Ref 21-24. Rapeseed oil as a bio-polyol source was reported much before the cited papers. Authors must give credit to pioneering researchers where its due.
Results and Discussion
Authors show "The microstructures of PUR materials in cross-sections parallel and perpendicular to the foam rise direction are shown in Table 1". This reviewer wishes to know at what depth the samples for cross-section measurement taken. Its well known that in spray foam, the outer layer would be almost open cell or combination of open+closed cell. But the images shown in Table 1 and Fig. 2 indicates 'cherry picking' of appropriate sites to show exclusively 'open cell'. A proper study on depth dependent morphology should be included in revised manuscript.
Table 4 has no units.
Authors say "Before testing, the samples were placed in a cryostat filled with liquid nitrogen for at least 5 minutes". Only 5 minutes !! This is too short. What does ASTM or DIN standards say about conditioning of polymer samples for cryogenic testing. Add this in revised manuscript.

There are many more problems like missing information: "Then an appropriate amount of isocyanate was added to the polyol premix" What is the "appropriate amount", be specific. What is "Rokopol"? More details regarding synthesis of "second one was a bio-polyol HF  (synthetized in the laboratory of Cracow University of Technology, Cracow, Poland)" must be added. Fig. 6 says "T". What is this "T"

Author Response

Thanks for the reviews. The authors tried to answer all your questions and the article was supplemented with additional information

Rev.2

  1. Abstract: Authors state "The article compares the properties of closed-cell PUR bio-foams produced on a laboratory scale and on an industrial scale" but the polyol sources used in laboratory PU foam and industrial PU foam is not the same. It's like comparing oranges with lemons though  both are in the citrus family. 

The biopolyol used for the synthesis of polyurethane foams produced on a laboratory and industrial scale (BIO_PUR and BIO_PUR_1:1) was the same.

  1. Introduction is very badly written and the cited papers are inappropriate. Authors say "full replacement of petrochemical polyols with bio-polyols is preferred [6–13]" and cite papers published in 2020, 2021 etc. Seriously do authors think that until 2020s no one else thought of replacing petroleum based polyols with biobased polyols? Also Ref 21-24. Rapeseed oil as a bio-polyol source was reported much before the cited papers. Authors must give credit to pioneering researchers where its due.

We add a few historical references – about the first review articles in topic of renewable materials in PU industry.
The literary introduction should be based on the most recent literature.
Of course, many researchers have certainly thought about carrying out research on scaling up spray foams synthesized with bio-polyols, but such studies have not been described so far.

  1. Results and Discussion
    Authors show "The microstructures of PUR materials in cross-sections parallel and perpendicular to the foam rise direction are shown in Table 1". This reviewer wishes to know at what depth the samples for cross-section measurement taken. Its well known that in spray foam, the outer layer would be almost open cell or combination of open+closed cell. But the images shown in Table 1 and Fig. 2 indicates 'cherry picking' of appropriate sites to show exclusively 'open cell'. A proper study on depth dependent morphology should be included in revised manuscript.

All samples for structure tests were taken from the center of the foam, the so-called core. Testing method has been completed. It is a commonly used research methodology. Spray foams can be open-cell or closed-cell structure. The materials described in this work have a closed-cell structure (closed-cell content> 85%), which was confirmed by tests, the results of which are presented in Table 2.

  1. Table 4 has no units.

Units have been completed.

  1. 5. Authors say "Before testing, the samples were placed in a cryostat filled with liquid nitrogen for at least 5 minutes". Only 5 minutes !! This is too short. What does ASTM or DIN standards say about conditioning of polymer samples for cryogenic testing. Add this in revised manuscript.

Yes, at least 5 min., because sample are soaked in LN2. But in real life ,  5 min wait, after that start “press buttons” on Zwick, and finally this conditioning period is ~10 min.

There are no standards for cryogenic compression tests, it is our self-designed test equipment.

  1. There are many more problems like missing information: "Then an appropriate amount of isocyanate was added to the polyol premix" What is the "appropriate amount", be specific. What is "Rokopol"? More details regarding synthesis of "second one was a bio-polyol HF  (synthetized in the laboratory of Cracow University of Technology, Cracow, Poland)" must be added. Fig. 6 says "T". What is this "T"

Rokopol 551 is a petrochemical polyol. The manufacturer is listed at the beginning of the Materials part. The exact characteristics are presented in Table 3. The text presents the method of HF bio-polyol synthesis and the characteristics of this bio-polyol is presented in Table 3.

Explanation of “T” has been added in text.

Round 2

Reviewer 1 Report

The authors have adequately addressed all of the comments, thus I am happy to recommend acceptance of this work for publication at Materials.